# Development and validation of a multigene variant profiling assay to guide targeted and immuno therapy selection in solid tumors

**Dadasaheb Akolkar** *⊕ ᵒ, **Darshana Patil** ᵒ, **Navin Srivastava** ᵒ, **Revati Patil, Vineet Datta, Sachin Apurwa, Nitin Yashwante, Raja Dhasarathan, Rahul Gosavi, Jinumary John, Shabishta Khan, Ninad Jadhav, Priti Mene, Dhanashri Ahire, Sushant Pawar, Harshal Bodke, Subhraline Sahoo, Arun Nile, Dinesh Saindane, Harshal Darokar, Pradip Devhare, Ajay Srinivasan, Rajan Datar**

Datar Cancer Genetics, Nashik, Maharashtra, India

⊙ These authors contributed equally to this work.
* dadasaheb.akolkar@datarpgx.com

## Abstract

We present data on analytical validation of the multigene variant profiling assay (CellDx) to provide actionable indications for selection of targeted and immune checkpoint inhibitor (ICI) therapy in solid tumors. CellDx includes Next Generation Sequencing (NGS) profiling of gene variants in a targeted 452-gene panel as well as status of total Tumor Mutation Burden (TMB), Microsatellite instability (MSI), Mismatch Repair (MMR) and Programmed Cell Death—Ligand 1 (PD-L1) respectively. Validation parameters included accuracy, sensitivity, specificity and reproducibility for detection of Single Nucleotide Alterations (SNAs), Copy Number Alterations (CNAs), Insertions and Deletions (Indels), Gene fusions, MSI and PDL1. Cumulative analytical sensitivity and specificity of the assay were 99.03 (95% CI: 96.54–99.88) and 99.23% (95% CI: 98.54% - 99.65%) respectively with 99.20% overall Accuracy (95% CI: 98.57% - 99.60%) and 99.7% Precision based on evaluation of 116 reference samples. The clinical performance of CellDx was evaluated in a subsequent analysis of 299 clinical samples where 861 unique mutations were detected of which 791 were oncogenic and 47 were actionable. Indications in MMR, MSI and TMB for selection of ICI therapies were also detected in the clinical samples. The high specificity, sensitivity, accuracy and reproducibility of the CellDx assay is suitable for clinical application for guiding selection of targeted and immunotherapy agents in patients with solid organ tumors.

## Introduction

The process of carcinogenesis traces back to the progressive accumulation of genomic alterations which lead to abnormalities in the genetic landscape, such as chromosomal and gene rearrangements, gene amplifications, deletions, aneuploidy, as well as loss-of-function or gain-of-function mutations [1]. Evaluation of these molecular landmarks of the malignancy in tumor tissue can provide crucial therapeutic guidance for selection of cancer-specific as well as

**Funding:** All author except RD2 are employees of the DCG, which is the Funding Institution and has commercial interests. The funding institution provided support in the form of salaries for authors [DA1, DP, NS, RP, VD, SA, NY, RD1, SK, NJ, PM, DA2, SP, HB, SS, AN, DS, PD, AS], but did not have any additional role in the study design, data collection and analysis, decision to publish, or preparation of the manuscript. The specific roles of these authors are articulated in the 'author contributions' section.

**Competing interests:** I have read the journal's policy and the authors of this manuscript have the following competing interests: All author except RD2 are employees of the Funding Institution, i.e., Datar Cancer Genetics (DCG), which offers commercial services in oncology. RD2 is the founder and Managing Director of the Funding Institution, and holds patents on products in development as well as marketed products which incorporate technology described in this manuscript as well as other similar technologies. This does not alter our adherence to PLOS ONE policies on sharing data and materials.

pan-cancer targeted and immune checkpoint inhibitor (ICI) therapies. Notable examples of each category include the selection of Epidermal Growth Factor Receptor–Tyrosine Kinase Inhibitors (EGFR-TKIs) such as Osimertinib in Lung cancer [2], selection of the pan-cancer drug Larotrectinib in malignancies which harbor NTRK fusions [3] or selection of the immune checkpoint inhibitor (ICI) Pembrolizumab in multiple cancers based on PD-L1 expression [4], microsatellite instability (MSI) or deficiency in Mismatch Repair (dMMR) genes [5] as well as tumor mutation burden (TMB) [6].

Sensitive and accurate detection of these molecular features is crucial for therapy selection in order to avoid risks of treatment failure owing to inaccurate assays. It is therefore equally imperative for stringent validations of molecular investigations before clinical adoption. Next generation sequencing (NGS) technology is a significant technological advancement which provides sensitive, accurate, high-throughput evaluation of variations in multiple genes and continues to evolve as a platform of choice for cancer diagnostic applications. The key advantages of NGS based gene profiling are the ability to simultaneously evaluate multiple (hundreds of) genes in the same run, in a short interval of time and with low requirement of DNA. Several NGS-based diagnostic assays find potential and actual applications in the clinical setting [7]. The United States Food and Drug Administration (US-FDA) have approved several NGS-based companion diagnostic assays which identify gene alterations to guide selection of targeted and ICI therapies [8–10].

In the present study, we report the validation of the CellDx tumor profiling assay, which includes NGS profiling of gene alterations (SNAs, CNAs, Indels, Gene Fusion and TMB), Capillary electrophoresis (CE) for Microsatellite instability (BAT-25, BAT-26, NR-21, NR-24, and MONO-27), IHC for detection of PD-L1 (28–8 and 22C3) expression and IHC for MMR status (MLH1, MSH2, MSH6, PMS2). Validation of the CellDx assay established the analytical and clinical sensitivity, specificity, reproducibility and limit of detection based on 122 samples and the real-world utility by a subsequent analysis of 299 samples from cancer patients.

## Materials and methods

### Samples and standards

A total of 421 samples were used for analytical validation and evaluation of real-world clinical performance of the CellDx assay. Analytical validation was performed on 122 reference samples including Formalin Fixed Paraffin Embedded (FFPE) tumor tissue, tumor DNA or tumor RNA (S1 Table) which were obtained from various sources such as College of American Pathologist (CAP), European Molecular Genetics Quality Network (EMQN), Coriell Institute of Medical Research (CIMR) as well as various commercial providers. For all reference and commercial samples, manufacturer's Certificate of Analysis was used as confirmation of sample characteristics. For Proficiency Testing (PT) samples obtained from CAP or EMQN, the sample specifications documents were used as confirmation. For samples with insufficient information such as clinical samples with previously detected variants, appropriate orthogonal testing was performed to ascertain the variant. Samples obtained were determined to be appropriate for each assay type such as NGS, MMR, MSI and PD-L1. In addition to the reference samples, 299 clinical specimens (S2 and S3 Tables) were obtained to assess the clinical performance of the assay.

### Ethics statement

All patients provided signed informed consent for the publication of deidentified data and results. The process of obtaining patients samples was in accordance with all regulatory and ethical guidelines including ICH-GCP and the Declaration of Helsinki. The use of patient

samples in this validation was approved by the Ethics Committee of the Study Sponsor (Datar Cancer Genetics Ethics Committee). Cellular and molecular investigations on the patient's samples were carried out at the College of American Pathologists (CAP)-accredited, Clinical Laboratory Improvement Amendments (CLIA)-accredited and International Organization for Standardization (ISO)-compliant facilities of the Study Sponsor.

## Assay designing and content

The workflow and overall components of CellDx assay was illustrated in **Fig 1**. Fresh tissue biopsy/ FFPE tissue samples were used to obtain 3–4 sections (5–10 μm) for DNA/RNA isolations and protein expressions. Mutations and gene fusions were detected in tissue samples by NGS using ion Ampliseq 452 gene panel on Ion S5 prime semiconductor sequencer (Thermo Fisher, USA). The Ion AmpliSeq 452 gene panel (Thermo Fisher, USA) with Ampliseq technology together with Ion proton and Ion S5 prime sequencer was selected and optimized for CellDx assay. The Ampliseq 452 panel was designed to analyze SNAs and Indels in 436 genes, CNAs in 417 genes and gene fusions in 51 gene. This gene panel comprises 139 oncogenes (OG), 110 tumor suppressor genes (TSG) and 40 genes known to have OG and TSG role. The TSG and OG annotation was based on data available in the Calalog of Somatic Mutations in Cancer database [11] (COSMIC: https://cancer.sanger.ac.uk/cosmic). Directly therapeutically targetable mutations were reported in 187 genes as per the My Cancer Genome [12] (https://www.mycancergenome.org/) database. Variants detected by NGS in 17 genes were indicative for selection of FDA-approved anticancer agents in labelled setting. In addition to variants in these 17 genes, variants in an additional 85 genes were also available for selection of anticancer agents in an off-label setting. Details of the gene panel are provided in S4 Table. The multigene panel was designed to include genes with high frequency of relevant actionable variants and prognostically significant variants, based on information available in public databases and literature. CellDx is not intended to predict whether variants are pathogenic, nor is it intended to annotate variants. CellDx is intended to provide such information available in public domain to assist clinicians in decision making. The NGS data was analyzed by torrent suite V 5.10 (Thermo Fisher, USA) and ion reporter V 5.10 (Thermo Fisher, USA) and variant calling file was uploaded to Ingenuity software version 5.6 (Qiagen, Germany) and PredictSNP2 to

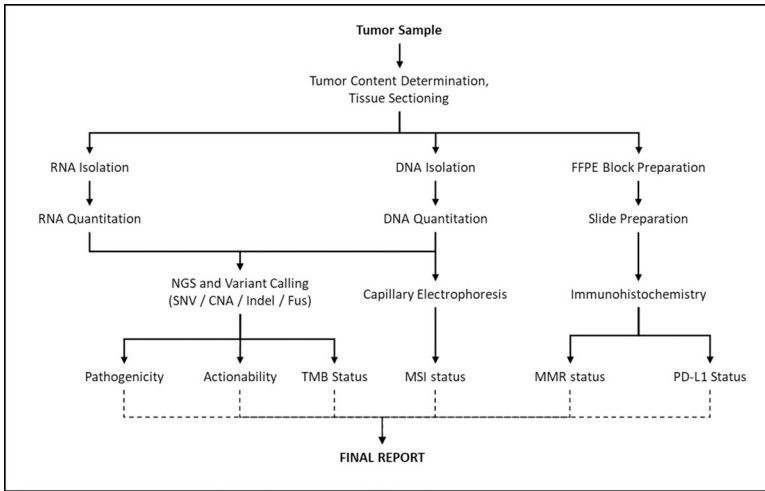

**Fig 1. CellDx assay workflow.** The schema depicts the sequence of sample processing and investigations in CellDx. CellDx requires freshly biopsied or formalin-fixed paraffin-embedded (FFPE) tumor tissue.

annotate the variants as either Pathogenic / Likely-Pathogenic / Driver / Passenger / Variant of Unknown Significance (VUS) etc. TMB was estimated (explained in a subsequent section) based on all somatic variants for selection of immune checkpoint inhibitors. Microsatellite instability was detected by capillary electrophoresis using the Promega MSI analysis system V2.1 (Promega, USA) for selection of immune checkpoint inhibitors. PD-L1, as well as MMR protein expressions were detected by Immunohistochemistry (IHC) for selection of immune checkpoint inhibitors. Results of NGS, MSI, PD-L1 and MMR was assigned to generate comprehensive clinical report.

## DNA/RNA extraction

Tumor content in FFPE tissue was determined by Hematoxylin and Eosin (HE) staining by a team of trained pathologists. Sections (4–5 of 10 μm each) with a minimum 5% tumor content were selected and used for nucleic acid isolations. FFPE DNA was isolated using Qiagen gene read FFPE DNA isolation kit (QIAGEN, Germany) and FFPE RNA was isolated using Maxwell Promega FFPE RNA/DNA isolation kit (Promega, USA) as per the manufacturer's instructions. After isolation quality of FFPE DNA/RNA was assessed using Nanodrop 2000 spectrophotometer (Thermo Fisher, USA). FFPE RNA was quantified using the Qubit fluorometer using the Qubit HS RNA assay kit (Thermo Fisher, USA) and FFPE DNA was quantified by RNase P assay (Thermo Fisher, USA) using 7500Dx Fast Real time PCR (Thermo Fisher, USA) as per manufactures instructions. Amplifiable FFPE DNA was calculated based on the quantity obtained by real time PCR.

## Library preparation and sequencing

The library preparation for each sample was done using the Ion Ampliseq 452 gene panel. The assay required 10–100 ng of RNA/DNA from FFPE tissue samples. For RNA samples cDNA was converted using the SuperScript™ VILO™ cDNA Synthesis Kit (Thermo Fisher, USA) followed by library preparation using Ion Ampliseq library kit plus (Thermo Fisher, USA) with 452 gene panel. The targets were amplified by thermal cycling as per manufacturer's recommendations by GeneAmp® PCR System 9700 (Applied Biosystems, USA). After target amplification PCR components were combined together and treated with the FuPa enzyme which were then ligated to multiplexing Ion Code barcodes (Thermo Fisher, USA). Libraries then purified by JetSeq™ Clean (Bioline, USA) magnetic beads and amplified using Amplification master mix (Thermofisher Scientific). Prepared Library underwent quality control (QC) using an E-Gel™ Agarose Gel 2% (Thermo Fisher, USA). Libraries were analyzed on Agilent 2100 bioanalyzer (Agilent Technologies, USA). Purified libraries were Quantified by Quant studio 12 k Flex Real Time PCR (Thermofisher, USA) using Ion library TaqMan quantification kit (Thermo Fisher, USA). The volume of each of the prepared libraries was diluted to 100 pmol to add equimolar concentration of each library into the emulsion PCR for a final total molarity ranging from 8 to 10 pM. The emulsion PCRs (Ion OneTouch™ 2) were carried out using Ion PI™ Hi-Q™ OT2 200 Kit. (Thermo Fisher, USA). After emulsion PCR non-templated Ion Sphere Particles (ISP) beads were enriched by streptavidin magnetic beads. After ISP bead enrichment, each library was sequenced using the Ion PI™ Hi-Q™ Sequencing 200 Kit (Thermo Fisher, USA). The enriched ISPs were loaded in Ion PI V3 Chip (Thermo Fisher, USA) and sequenced on Ion Proton semi-Conductor Sequencer, which acquired sequencing data points and generated a BAM and a FASTQ files.

## Sequencing data analysis

Raw data analysis was performed using torrent suite software version 5.10 (Thermo Fisher, USA) along with ion reporter version 5.10 (Thermo Fisher, USA) by default analysis

parameters. For DNA sequencing raw reads were aligned to human genome 19 using the Torrent Mapping Alignment Program (TMAP) plug in with default parameters and variant calling was performed using the Variant Caller version 5.10 using the torrent variant caller (TVC) plug in (Thermo Fisher, USA). A minimum sequencing depth of 500X with an allelic frequency of 2.5% was used as a cutoff with at least 20 variants reads to be called a variant. All fusions with read counts ≥120 were considered as positive. Ingenuity Variant Analysis software version 5.6 (Qiagen, Germany) was used for annotation and Ion reporter version 5.10 (Thermo Fisher, USA) was used for copy number and gene fusion detection in their respective workflow as per default parameters. For clinical samples, all detected variants were mapped to CIViC [13] (https://civicdb.org/) and OncoKB [14] (https://oncokb.org/) clinical annotation databases.

## Tumor mutational burden from NGS data

Tumor mutational burden (TMB) was determined by NGS using ion Ampliseq 452 gene panel in 133 FFPE/fresh tissue samples which were available in sufficient quantity. Minimum sequencing depth was 500X. TMB was defined as the sum total of somatic mutations, and calculated as the count of all non-synonymous, 5' and 3' splice-site variants present at ≥5%. Raw counts of AF were queried against University of California Santa Cruz (UCSC) database of common Single Nucleotide Polymorphisms (SNPs), Exome Aggregation Consortium (ExAC) database, 5000 Exomes database and dbSNP to filter common variants. For each cancer type, TMBs were classified as Low, Intermediate or High based on thresholds defined in literature [15–22]; Low was 1–6 mut/MB in Lung Cancers, 1–9 mut/MB in Breast Cancers and 1–10 mut/MB for all other solid tumors; Intermediate was 6–20 mut/MB in Lung Cancers, 9–20 mut/MB in Breast Cancers and 11–20 mut/MB for other solid tumors; High was >20 mut/MB in all cancer types.

## Microsatellite Instability (MSI) by Capillary Electrophoresis (CE)

MSI was evaluated in 4 samples by multiplex PCR using fluorophore-tagged primers for five mononucleotides repeat markers (NR-21, BAT-26, BAT-25, NR-24, MONO-27) using the MSI Analysis kit, Version 1.2 (Promega, USA), as per the manufacturer's instructions. PCR run was performed in the GeneAmp® System 9700 Thermal Cycler. Reactions were set up in 96- well plates, which were centrifuged briefly (1500×g for 2 min) to remove air bubbles. Samples were denatured (95˚C, 3 min), immediately transferred to a cooling block and then loaded in ABI 3500Dx instrument (Applied Biosystems, USA) for CE (Promega, USA). Data was analyzed using GenMapper® Software version 5 (Applied Biosystems, USA). Results were interpreted as MSI-High (≥ 2 unstable), MSI-Low (1 unstable) or MSS (Microsatellite Stable, no unstable markers).

## Mismatch repair (MMR) by Immunohistochemistry (IHC)

A panel of ready to use antibodies (DAKO EnVision FLEX primary mAb, anti-MLH1 clone-ES05, anti-MSH2 clone-FE11; anti-MSH6 clone-EP49; anti-PMS2 clone-EP51) were used to determine MMR protein expression in 10 samples. FFPE tissue blocks were used to prepare 3–4 μm tissue sections on poly-L lysine coated slides (Leica, Germany) which was placed on hot plate at 60˚C for 1 hour. Deparaffinization, rehydration, antigen retrieval, antigen blocking and staining were performed in an automated slide staining system (Leica, Germany) as per the manufacturer's instructions. Each IHC run contained a positive control. Post-IHC, the slides were dehydrated and mounted using Distyrene, Plasticizer, Xylene (DPX) mountant. Results were interpreted by an experienced pathologist under light microscopy (Leica,

Germany). Intact nuclear staining was considered as no loss of nuclear expression (NLNE) whereas absolute absence of nuclear staining was considered as loss of nuclear expression (LNE). MMR protein expression was grouped into five categories: NLNE, LNE: MLH1/PMS2, LNE: MHS2/MSH6, LNE: MSH6 and LNE: PMS2. Since MMR deficiency leads to MSI, evaluation of MSI by CE and MMR by IHC were considered interchangeable investigations.

## PD-L1 by Immunohistochemistry (IHC)

A panel of antibodies (Dako, PD-L1 IHC 22C3; pharmDx and PD-L1 IHC 28–8; pharmDx) were used to determine PD-L1 expression in 40 samples. FFPE tissue blocks were used to obtain 3–4 μm sections of tumor tissue on poly-L lysine coated slides (Leica, Germany) which was placed on hot plate at 60˚C for 1 hour. Deparaffinization, rehydration, antigen retrieval, antigen blocking and staining were performed in an automated slide staining system (Leica, Germany) as per the manufacturer's instructions. Each IHC run contained a positive control. Post-IHC, the slides were dehydrated and mounted using Distyrene, Plasticizer, Xylene (DPX) mountant. Results were interpreted by an experienced pathologist under light microscopy (Leica, Germany). Based on the observations, the tumor proportion score (TPS) was obtained as the fraction (%) of intact tumor cells with partial or complete membrane staining. TPS score ≥ 1% was considered as PD-L1 positive.

## Assay sensitivity and specificity

The analytical sensitivity was defined as the ability of the CellDx assay to detect known variants in NGS as well as MSI, MMR and PD-L1 statuses. A total 110 samples were used to calculate the sensitivity and Specificity of CellDx test, including 56 samples for NGS (149 SNAs, 13 small indels, 4 large indels, 2 CNA, and 10 gene fusions), 20 positive samples for PD-L1 (10 samples for clone 28–8 and 10 samples for clone 22C3), 20 negative samples for PD-L1, 10 samples for MMR (MLH-1 and PMS2) and 4 samples for MSI. The analytical sensitivity and specificity was calculated for SNA, CNAs, large indels (> 4bp), small indels (< 4bp), gene fusions, MSI, MMR and PD-L1. The observed result was compared with expected (known) results for each sample and indicated as either a true-positive (TP, expected variant detected) or a false-negative (FN, expected variant not detected). The analytical sensitivity was calculated by determining the fraction of TP among the sum of TP and FN [TP/ (TP+FN)] and the 95% CI was estimated using Medcalc diagnostic calculator (https://www.medcalc.org/calc/diagnostic_test.php). The acceptable sensitivity was prespecified as ≥ 95% [23–25]. Similarly, the analytical specificity was obtained as the fraction of true negatives (TN, expected wild type detected = no unexpected variants detected) among the sum total of TN and false positive (FP, expected wild type not detected = unexpected variant detected), [TN / (TN + FP)]. The 95% CI was estimated using Medcalc's diagnostic test evaluation calculator. The target for acceptable specificity was ≥ 98%. The acceptable mean number of false-positive results per tested sample was prespecified to be ≤ 2% [23–25].

## Accuracy

Accuracy [23] was assessed from sensitivity (TP and FN) and specificity (TN and FP) assessments from 110 samples. Accuracy was defined as the proportion (%) of TP and TN among the sum total of TP, TN, FP and FN. The 95% CI was estimated using the Medcalc's diagnostic test evaluation calculator.

## Assay precision

The Repeatability and Reproducibility of the CellDx assay was evaluated using 31 samples. Two operators processed the same samples independently and each operator performed each assay twice. Repeatability and Reproducibility were assessed for Ampliseq 452 gene panel, MMR, MSI and PD-L1 Precision was assessed by calculating positive concordances between pairwise inter and intra-user comparisons. Positive pairwise concordance was defined as the fraction of positive results in agreement among the total positive results between the replicates [23–25].

## Preliminary clinical feasibility study

Clinical feasibility of the CellDx assay was explored in a preliminary study based on evaluation of patient derived tumor samples (fresh tissue or FFPE blocks) for detection of actionable features in DNA/RNA, as well as in determining status of TMB, MMR / MSI and PD-L1. The study intended to determine the prevalence of clinically significant and actionable variants in patient samples. This preliminary study was based on remnant archived patient samples with appropriate consent from patients to use deidentified samples for research, development and validation purposes as well as for publication of sample-derived data. No patient underwent any additional invasive procedure to obtain samples for this Study. FFPE tissue samples, fresh tissue samples or 4–5 core needle biopsy specimens from 299 cancer patients were used and processed as per the protocol mentioned above.

# Results

## NGS data analysis

Nucleic acid isolated from 20 tissue types, PT(CAP) and EMQN samples and commercial samples were used for library preparation. Library yields for all DNA and RNA samples exceeded the minimum requirements of 100 pmol/L irrespective of sample types indicating that the method was compatible for all type of samples.

NGS data obtained for 83 samples from 11 sequencing runs of the 452 gene panel were used for analytical validation. The 452 gene panel produced a median of 8,808,657 reads per sample (range, 1.45 M to 30.33 M), a median read length of 111 bp (range, 90 to 125 bp), 94.0% median on target reads and a median 96.0% uniformity of data. Uniformity was defined as the proportion (%) of target bases covered by at least 0.2x the mean base-read depth [26]. By targeting >500X mean depth from the analysis of all samples across 11 runs, 99.1% amplicons were covered in the 452-gene panel coverage at >100X (**Fig 2**). NGS Run Statistics are provided in S5 Table. Sample-wise variants are provided in S6 Table.

## Limit of detection for NGS

The LOD of the assay was based on two parameters, minimum tumor content and the lower VAF in tumor DNA. Two FFPE tissue samples were used for evaluating LOD for 4 types of variants, i.e., CNA, SNV, InDel and Fusion. Tumor content of the tissue samples were determined by a pathologist and DNA / RNA were isolated. To evaluate the LOD of variant calling, DNA was serially (2-fold) diluted with wild type CEPH DNA to simulate 70%, 35%, 17.5% and 8% tumor content and estimate the lowest VAF for 3 variant types, i.e., SNA, CNA and Indel.

DNA obtained from tissue with 70% tumor content was known to harbor 37 copies of CCNE1 (CNA), TSC1_c.2065C>T at 47% (SNA) and FANCI_ c.1641_1642delTA (small indel) at 3.7% VAF. The default parameters for LOD assessments were 2.5% VAF detection threshold and 20 sequencing reads with the variant was not applied. Across the serially diluted

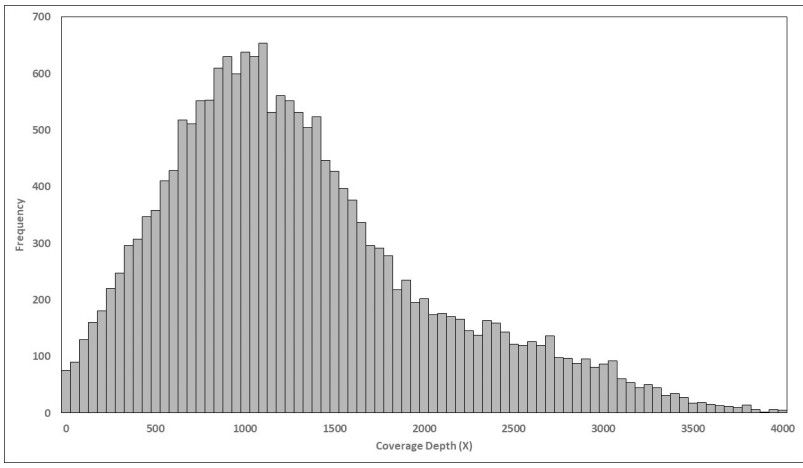

**Fig 2. Distribution chart of amplicon coverage in the targeted NGS panel.** The graph summarizes pooled performance of 83 samples over 11 NGS runs. There were 8,808,657 (median) reads per sample, with read length of 111 bp (median). The mean targeted depth was >500X, and the majority (99.1%) of amplicons were covered at >100X.

samples, VAFs of TSC1_c.2065C>T SNA ranged from 47.00% to 1.27%, CNA for CCNE1 (first 3 dilutions) ranged from 37 to 4. Non-linearity in VAF was observed for FANCI c.1641_1642delTA indel dilutions (3.7 to 0.9%) and are speculated to arise from the proportion of wt DNA reads in this region.

Similarly, RNA obtained from tumor tissue with 60% tumor content was known to have 2028 fusion copies of EML4-ALK. The RNA was serially (2-fold) diluted with healthy RNA sample to estimate limit of gene fusion variant. Gene fusions were detected from 2028 to 298 read counts. The LOD observations were consistent with the default detection limits of the data analysis pipeline. Based on these observations, minimum tumor content (MTC) was defined as ≥ 8% for SNA and indels, and ≥ 25% for CNA and gene fusion respectively at VAF ≥4% (Table C, in S7 Table). Based on evaluation of LOD the threshold for calling SNAs and Indels were set at 2.5% VAF for a sample with 8% tumor content, for calling CNAs were set at gain of 3 copies or loss of 1 copy for a sample with 25% tumor content, and for calling of fusions minimum read count was set at >120 with for a sample with 25% tumor content. Significant, i.e., therapeutically actionable, variants with VAF <2.5% are reviewed by ddPCR for positive or negative.

## Analytical sensitivity and specificity

Analytical Sensitivity and Specificity of the CellDx assay was determined on 110 samples. Among the 56 samples which were analyzed by NGS, 1286 variants were detected in 289 genes including 1211 SNA, 34 indels (26 small and 8 large), 10 CNA and 31 gene fusions. Review of BAM files using integrated genome viewer [27] indicated that among these 1286 variants, 1284 variants were above the LOD threshold for respective Variant Allele Frequencies (VAF) and 2 variants were below the LOD (EGFR.pL858R at 2% AF and MET at 2.8 copies).

All 1284 (100% of 1284, or 99.85% of 1286) were accurately detected and called by the Ion reporter software V5.10. The 2 variants at below threshold VAF were reported as False Negatives. The VAF ranged from 3.1% - 100% for SNAs; 3.0% - 39.0% for small indels; 2.6% - 55.65% for large indels, while fusion reads were 67 to 191,644 and 11 copies for CNAs (MYC-N).

With the exception of 9 false positive (FP) variants, 1099 true negative (TN) variants were correctly detected by NGS which indicated a Specificity of 99.19%.

**Table 1. Analytical validation of the CellDx assay.**

| Analytes | Sensitivity % (95% CI) | Specificity % (95% CI) | Accuracy % (95% CI) | Precision % |
|---|---|---|---|---|
| NGS: SNA | 99.33 (96.32–99.98) | 99.15 (98.40–99.61) | 99.17 (98.49–99.60) | 100.00 |
| NGS: Small Indels | 100.00 (75.29–100) | 100.00 (75.29–100) | 100.00 (86.77–100) | 100.00 |
| NGS: Large Indels | 100.00 (39.76–100.00) | 100.00 (39.76–100.00) | 100.00 (63.06–100) | |
| NGS: Fusions | 100.00 (69.15–100) | 100.00 (83.89–100) | 100.00 (88.78–100) | 90.00 |
| NGS: CNA | 50.00 (1.26–98.74) | 100.00 (63.06–100) | 90.00 (55.50–99.75) | 100.00 |
| IHC: PD-L1 | 100.00 (83.16–100) | 100.00 (83.16–100) | 100.00 (91.19–100) | 100.00 |
| IHC: MMR | 100.00 (15.81–100) | 100.00 (89.11–100) | 100.00 (89.72–100) | 100.00 |
| IHC: MSI | 100.00 (54.07–100) | 100.00 (69.15–100) | 100.00 (79.41–100) | 100.00 |
| **Overall** | **99.03 (96.54–99.88)** | **99.23 (98.54–99.65)** | **99.20 (98.57–99.60)** | **99.70%** |

Validation Parameters for each of the individual assays with performance scores, and 95% CI. Expanded details including number of samples, number of variants, variants called / detected (True Positives (TP), True Negatives (TN), False Positives (FP) and False Negatives (FN)) for determination of Sensitivity, Specificity and Accuracy, replicate observations for determining Precision, and dilutions for determining Linearity are provided in S7 Table.

Sensitivity and specificity were also determined for MSI, PD-L1 and MMR (N = 54 samples). Sensitivity and Specificity were 100.00% each for MSI, PD-L1 and MMR respectively. The overall sensitivity of the CellDx assay was 99.03% (95% CI: 96.54% - 99.88%) and the overall specificity was 99.23% (95% CI: 98.54% - 99.65%) (Table 1; S7 Table).

## Accuracy

Accuracy of CellDx was determined from analysis of 110 samples which included 51 known positives for Sensitivity and 59 known negatives for Specificity. Overall accuracy of CellDx assay was based on detection of all TP variants (N = 204), TN variants (N = 1161), FP variants (N = 9) and FN variants (N = 2). The overall accuracy of the CellDx assay was 99.20% (95% CI: 98.57% - 99.60%). (Table 1; S7 Table). The Accuracy of each investigation is also provided in S7 Table.

## Precision (repeatability and reproducibility)

Precision including Intra-Operator Concordance (Repeatability) and Inter-Operator Concordance (Reproducibility) were determined for NGS, PD-L1, MMR and MSI. Precision was determined from 31 known positive samples, of which 9 were evaluated for NGS, 6 were evaluated by MSI, 4 were evaluated for MMR and 12 were evaluated for PD-L1 (IHC). For NGS replicates included library preparation, sequencing and data analysis. For PD-L1 and MMR, replicates encompassed sample preparation to staining and interpretation of results. For MSI replicates encompassed DNA isolation to interpretation of results. For NGS, 276 variants were evaluated.

MSI STR markers were successfully detected in both replicates as well as MMR and PD-L1 analysis was reproducible. Inability to detect a fusion (SLC34A2-ROS1) in one replicate translated into 99.7% overall concordance (Table 1; S7 Table).

## Preliminary clinical feasibility

The CellDx assay was used to evaluate 299 tissue samples from patients with known cases of cancer to determine molecular variants by NGS as well as TMB, MSI / MMR and PD-L1 status.

Among the 133 overall samples which were profiled by NGS samples, significant somatic variants (pathogenic, likely pathogenic, VUS) were detected in 96.99% (129/133) of samples. A total of 4,666 reportable variants were detected including 3,355 CNAs (37% loss; 63% gain), 1,161 SNA, 129 indels and 21 gene fusions. In all 852 unique mutations were detected, of which 784 were oncogenic and 47 were actionable, i.e., indication for selection of a targeted anticancer drug (S8 Table). All reported variants had a median VAF of 20.15%, and 99.98% variants were detected at $\geq$ 4.0% AF. The most frequently detected alterations were in *TP53* (25%) and *PIK3CA* (7%). Gene fusions detected in 7% patients (21/299), mostly included *EIF3E*, *NCOA4*, *PTPRK*, *ESR1*, *FGFR3*, and *MYB*.

Among the 133 samples considered for TMB, 5 (3.8%) were excluded due to deamination induced interference. In the remaining 128 patient samples, TMB was determined to be in the range of 0–88.27 mut/MB with a median of 7.6 mut/MB. TMB levels was divided into three groups for each cancer based on available literature [11–18]. Among the 128 patients, 83 patients (65%) had low TMB, 32 patients (25%) had an intermediate TMB and 13 patients (10%) had high TMB. Higher TMB was more frequently encountered in cancers of the Breast, Cervix, Colorectum, Oesophagus, Liver Stomach and Lung as well as in Sarcomas.

Among the 105 samples evaluated for MMR status, LNE: MSH2 + MSH6 was observed in 1 sample, indicating potential benefit from Immune Checkpoint Inhibitor (ICI) therapies. Among the 138 samples evaluated for MSI status, 6 samples (4%) returned positive findings with BAT-26 (5%), NR-21 (2%), NR-24(2%) and MONO-27 (1%) indicating potential benefit from (ICI) therapies. Among 239 unique samples that were evaluated for both MMR and MSI, actionable indication was observed in 7 (2.9%) samples. Among the 112 total samples evaluated by IHC for PD-L1 expression, 22 samples (19.64%) were positive (TPS score >1%) for PD-L1 indicating that these patients were likely to respond favorably to treatment with ICI Therapies. Overall, 73 patients were eligible for targeted or immune-therapy agents in labelled indication, 29 patients were eligible for targeted or immune-therapy agents in an off-label setting, while 19 patients had both types of indications (labelled and off-label).

Among the 57 clinical samples in which all evaluations (NGS, TMB, MMR/MSI and PD-L1) were performed, 37 samples had actionable indications, which included 19 samples with variants indicative for selection of a targeted anticancer drug, 21 samples with high or intermediate TMB, 1 sample positive for MMR / MSI and 12 samples positive for PD-L1.

## Discussion

Precision Oncology aims to provide personalized treatment options based on patient-derived *de novo* evidence which can be obtained from multi-analyte, multi-variant evaluation of the tumor. A prior single institution retrospective study has shown that tumors in majority of patients have biologically actionable alterations [28], of which several can be therapeutically targeted with approved agents, while others are in various phases of clinical trials. It is well accepted that multigene variant profiling of tumor tissue samples can guide selection of targeted and ICI therapies [29–34] as well as predict individuals who are more likely to respond to (or not respond to) systemic anticancer therapies. While several companion and complementary diagnostics assays have been approved by the US FDA [10] which guide selection of anticancer agents or predict likely responders, these are based on univariate analysis and for a single drug The CellDx assay on the other hand is a multi-gene variant profiling that provides a comprehensive profiling of actionable and therapeutically relevant tumor vulnerabilities.

The CellDx assay was developed and has been validated for detection of SNAs, CNAs, Indels, gene fusion by NGS, status of MSI by CE and status of PD-L1 as well as MMR by IHC. The high overall analytical sensitivity (99.03%) of the CellDx assay resulted from high

individual sensitivities of NGS, IHC and CE for all variants tested. The high sensitivity implies a little or no risk, if any, of undetectable actionable variants. The CellDx assay demonstrated an overall 99.23% specificity, as well as an accuracy of 99.20% and 99.7% precision indicating suitability for clinical use.

The clinical feasibility of the CellDx was explored using 299 clinical samples to identify molecular features on NGS as well as MSI, PD-L1 and MMR status. Actionable findings in the 299 clinical samples were conveyed to the respective clinicians, but the patients were not followed up with to determine whether the findings were used to guide further therapeutic directions or evaluate treatment outcomes–these aspects were beyond the scope of the present manuscript.

Accurate detection of prognostic and predictive biomarkers can identify patients more likely to benefit from targeted and ICI therapies. The use of gene profiling for selection of targeted anticancer therapies based on gene variants is already well accepted. In addition, profiling of PD-L1, MMR and TMB are more recently developed biomarkers which guide ICI therapy selection. Targeting the immune checkpoint proteins (PD-L1 or PD1) with inhibitory mABs is a treatment strategy in multiple cancers [32, 33]. The expression of PD-1 and PD-L1 proteins was considered to be associated with response rate to ICI [29, 32, 33], and immunohistochemistry (IHC) profiling of PD-L1 status is routinely used to identify patients likely to benefit from ICI therapies [34]. More recent studies appear to indicate elevated TMB rather than PD-L1 expression as a more accurate predictor of treatment response [32–35]. Prior studies have also shown the association of tumors deficient for mismatch repair (dMMR) with higher response to ICI therapies [36]. The status of 4 MMR proteins, MLH1, MSH2, MSH6, and PMS2, as well as the LNE leading to dMMR status [30, 31] due to either germline or somatic mutation or inactivation by hypermethylation is determined by IHC [32]. dMMR status is associated with the accumulation of mutations in microsatellite regions, leading to microsatellite instability (MSI). Somatic mutations leading to dMMR have been shown to be associated with increased TMB and MSI [33–37], the latter being another predictive biomarker for response to ICI therapies [35].

In a routine setting, patients usually undergo evaluation of single variants at each instance for therapy selection, e.g., in NSCLC, evaluation of EGFR mutations, ALK-fusions, PD-L1 and MMR status are usually not performed simultaneously which leads to extended time to treatment. Similarly, in CRC, evaluation of RAS mutations, PD-L1, MMR and MSI are not performed simultaneously in the routine clinical setting. CellDx is advantageous in evaluating all variants and indicated therapy options at the outset which can enable appropriate therapy selection. CellDx may be perceived as a comprehensive companion and complementary diagnostics solution for selection of anticancer agents in labeled as well as label-agnostic settings; the latter is especially helpful for clinicians who are considering clinician's choice of treatments in refractory cancers where Standard of Care (SoC) treatment options are exhausted or unviable. Findings of CellDx not only directly identify treatment options, but also help to stratify patients as likely to respond and less likely to respond; the former is significant in guiding treatment selection in labelled as well as label-agnostic setting, while the latter can help avoid selection of futile treatments which yield not benefit to patient but add to the cumulative toxicity.

In summary, the ability of the CellDx assay to detect and report actionable variants in via NGS (for SNAs, CNAs, small indels, large indels and gene fusions) as well as to determine status of TMB, MSI, MMR and PD-L1 has direct and significant clinical utility in cancer management. The study findings sufficiently establish the high sensitivity, specificity, accuracy and reproducibility which is expected for clinical adoption of this assay.

## Supporting information

**S1 Table. Sample details used in analytical validation.**
(XLSX)

**S2 Table. Clinical sample details.**
(XLSX)

**S3 Table. Clinical sample demographics.**
(XLSX)

**S4 Table. Next Generation Sequencing (NGS) targeted 452 gene panel.**
(XLSX)

**S5 Table. NGS run statistics.**
(XLSX)

**S6 Table. Sample-wise distribution of variants.**
(XLSX)

**S7 Table. Performance characteristics of CellDx assay.**
(XLSX)

**S8 Table. Clinical actionability.**
(XLSX)

## Acknowledgments

The authors are grateful towards all patients who provided tumor tissue samples and consented for research, development and validation purposes, as well as for publication of deidentified data. The contributions of other members from the Sponsor Institute (DCGL) towards managing various clinical, operational and laboratory aspects of the study are acknowledged with gratitude.

## Author Contributions

**Conceptualization:** Dadasaheb Akolkar, Darshana Patil, Navin Srivastava, Rajan Datar.

**Data curation:** Dadasaheb Akolkar, Darshana Patil, Navin Srivastava, Revati Patil, Sachin Apurwa, Raja Dhasarathan, Rahul Gosavi, Jinumary John, Shabishta Khan, Ninad Jadhav, Priti Mene, Dhanashri Ahire, Sushant Pawar, Harshal Bodke, Subhraline Sahoo, Arun Nile, Dinesh Saindane, Ajay Srinivasan.

**Formal analysis:** Dadasaheb Akolkar, Darshana Patil, Navin Srivastava, Revati Patil, Sachin Apurwa, Nitin Yashwante, Raja Dhasarathan, Rahul Gosavi, Jinumary John, Shabishta Khan, Sushant Pawar, Harshal Bodke, Dinesh Saindane, Harshal Darokar, Pradip Devhare.

**Methodology:** Dadasaheb Akolkar, Darshana Patil, Navin Srivastava, Revati Patil, Nitin Yashwante, Rahul Gosavi, Jinumary John, Ninad Jadhav, Priti Mene, Dhanashri Ahire, Subhraline Sahoo, Arun Nile.

**Project administration:** Dadasaheb Akolkar, Darshana Patil, Navin Srivastava, Vineet Datta.

**Resources:** Rajan Datar.

**Software:** Sachin Apurwa.

**Supervision:** Raja Dhasarathan, Shabishta Khan, Dinesh Saindane, Rajan Datar.

**Validation:** Revati Patil, Sachin Apurwa, Ninad Jadhav, Priti Mene, Dhanashri Ahire, Harshal Bodke, Subhraline Sahoo, Arun Nile.

**Visualization:** Rahul Gosavi, Jinumary John.

**Writing – original draft:** Dadasaheb Akolkar, Darshana Patil, Navin Srivastava, Nitin Yashwante, Ajay Srinivasan.

**Writing – review & editing:** Dadasaheb Akolkar, Darshana Patil, Navin Srivastava, Revati Patil, Vineet Datta, Pradip Devhare, Ajay Srinivasan, Rajan Datar.

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
