## [Decision Letter · Decision Letter 0]

30 Oct 2020

PONE-D-20-23324

Development and Validation of a Multigene Variant Profiling Assay to Guide Targeted Therapy Selection in Solid tumors

PLOS ONE

Dear Dr. Akolkar,

Thank you for submitting your manuscript to PLOS ONE. We regret the delay in returning this decision and appreciate your patience.  After careful consideration, we feel that it has merit but does not meet PLOS ONE’s publication criteria as it currently stands. Therefore, we invite you to submit a significantly revised version of the manuscript that addresses the points raised during the review process.

Both reviewers have raised significant concerns about the the statistical analyses used as well as the specificity, sensitivity and reproducibility of the assay, with additional details needed in several instances.  As noted in the critiques, there also are reviewer comments that should be addressed related to the rationale for selecting the genes to profile and for using a subset of samples for particular analyses.  Reviewer 2 additionally raises important points about the underlying biology, highlighting key issues related to how pathogenicity and oncogenecity are defined, the differences between somatic and germline variants, and the clinical significance of the model.

We look forward to receiving your revised manuscript.

Kind regards,

Robert S. Weiss

Academic Editor

PLOS ONE

Journal Requirements:

2.Thank you for including your ethics statement: 

"The patient provided signed informed consent for the publication of deidentified data and results. The process of obtaining patients samples was in accordance with all regulatory and ethical guidelines. The use of patient samples in this validation was approved by the ethics committees of the sponsor as well as all participating institutes. Cellular and molecular investigations on the patient’s samples were carried out at the College of American Pathologists (CAP)-accredited and International Organization for Standardization (ISO)-compliant facilities of Datar Cancer Genetics (DCG), Nasik, India.".   

3.Thank you for stating the following in the Competing Interests section:

[I have read the journal's policy and the authors of this manuscript have the following competing interests: All author except Rajan Datar are employees of Datar Cancer Genetics (DCG), which offers commercial services in oncology. Rajan Datar is the founder and Managing Director of DCG. ].   

We note that one or more of the authors are employed by a commercial company: Datar Cancer Genetics

Reviewers' comments:

Reviewer's Responses to Questions

**Comments to the Author**

1. Is the manuscript technically sound, and do the data support the conclusions?

Reviewer #1: No

Reviewer #2: Partly

2. Has the statistical analysis been performed appropriately and rigorously? 

Reviewer #1: No

Reviewer #2: Yes

3. Have the authors made all data underlying the findings in their manuscript fully available?

Reviewer #1: No

Reviewer #2: Yes

4. Is the manuscript presented in an intelligible fashion and written in standard English?

Reviewer #1: Yes

Reviewer #2: Yes

5. Review Comments to the Author

Reviewer #1: The authors aimed to validate a multigene variant profiling assay to guide cancer targeted therapy. The assay includes a 452-gene next-generation sequencing target panel, microsatellite instability, tumor mutation burden and PD-L1 immunohistochemistry. The authors performed assay validation using 116 reference samples with known variants and 299 clinical samples. The validation of assay performance is important for reliable molecular diagnosis to guide cancer therapy. My major concerns are about the missing of key evaluation metrics, and the lack of comprehensible statistical analyses.

1. How was the uniformity defined? There was only mentioning of uniformity in Results with a median of 96% uniformity (Line 261). This is a key metric in the evaluation of the assay performance. How were the uniformities of the targeted 452 genes distributed among the samples?

2. The authors described ‘Most amplicons in the 452-gene panel were covered at an average read depth of…’ Exactly how many amplicons is an important metric to evaluate the assay performance, and should be calculable and presented, instead of describing as ‘most’.

3. The sensitivities and specificities of various tests were overly simplified in Results and Table. I cannot understand how the statistics were calculated. How could the authors get 99.99% sensitivity, given that there were two false negative mutations out of a total 176 variants (?) as described in the Results?

4. The cut-off values for calling SNVs appeared to be inconsistent, 4% in Line 156 and 2.5% in Line 275.

Reviewer #2: The authors present a multigene variant profiling assay (CellDx) and validate its high sensitivity using known variants in reference samples. However, the specificity and accuracy of this assay need further validation. The definition of true negative variants needs to be clarified. Furthermore, the oncogenicity of the detected variants and whether they are biologically actionable are annotated by external databases in this manuscript. CellDx alone is not sufficient to provide actionable indications for targeted therapy. In addition, whether and how the detected variants could be used as biomarkers to stratify patients also needs to be elaborated. More evidence is needed to support the clinical significance of CellDx.

Please see attachment to the decision letter for additional comments.

6. PLOS authors have the option to publish the peer review history of their article (what does this mean?). If published, this will include your full peer review and any attached files.

Reviewer #1: No

Reviewer #2: No

---

## [Author Response · Author response to Decision Letter 0]

11 Dec 2020

Reviewer #1: 

The authors aimed to validate a multigene variant profiling assay to guide cancer targeted therapy. The assay includes a 452-gene next-generation sequencing target panel, microsatellite instability, tumor mutation burden and PD-L1 immunohistochemistry. The authors performed assay validation using 116 reference samples with known variants and 299 clinical samples. The validation of assay performance is important for reliable molecular diagnosis to guide cancer therapy. 

My major concerns are about the missing of key evaluation metrics, and the lack of comprehensible statistical analyses.

1. How was the uniformity defined? There was only mentioning of uniformity in Results with a median of 96% uniformity (Line 261). This is a key metric in the evaluation of the assay performance. How were the uniformities of the targeted 452 genes distributed among the samples?

Author Response: Uniformity was defined as the proportion (%) of target bases covered by at least 0.2x the mean base-read depth. This definition is now included in the results section. Sample-wise uniformity data is provided in Supplementary Table S5.

2. The authors described ‘Most amplicons in the 452-gene panel were covered at an average read depth of…’ Exactly how many amplicons is an important metric to evaluate the assay performance, and should be calculable and presented, instead of describing as ‘most’.

Author Response: The statement has been updated for improved clarity and accuracy.

3. The sensitivities and specificities of various tests were overly simplified in Results and Table. I cannot understand how the statistics were calculated. How could the authors get 99.99% sensitivity, given that there were two false negative mutations out of a total 176 variants (?) as described in the Results?

Author Response: Table 1 provides the aggregated performance data for all parameters, while Supplementary Table S7 (which was previously Supplementary Table 5) provides comprehensive details on number of samples, number of variants, the true positives (TP), false positives (FP), true negatives (TN) and false negatives (FN) which are used for determination of Sensitivity, Specificity and Accuracy. 

As suggested by Reviewer 2, the initially separate Sensitivity and Specificity datasets were merged and cumulatively used to recalculate Sensitivity, Specificity and Accuracy. The previously determined values have been replaced with updated values in the Results section. The calculation is also explained in the Results section based on values in Supplementary Table S7. 

4. The cut-off values for calling SNVs appeared to be inconsistent, 4% in Line 156 and 2.5% in Line 275.

We regret the inadvertent error which has been rectified. The true cut off value is 2.5%.

Reviewer #2: 

The authors present a multigene variant profiling assay (CellDx) and validate its high sensitivity using known variants in reference samples. 

1. However, the specificity and accuracy of this assay need further validation.

Author Response: As suggested by the reviewer in a subsequent section, the Sensitivity, Specificity and Accuracy have been re-evaluated by merging the previously separate Sensitivity and Specificity data-sets. This has increased the overall sample-size and adds to the reliability of the data.

2. The definition of true negative variants needs to be clarified.

Author Response: The definition of True Negative variants has been clarified.

3. Furthermore, the oncogenicity of the detected variants and whether they are biologically actionable are annotated by external databases in this manuscript.

Author Response: CellDx is not intended for annotation of detected variants. Rather, CellDx retrieves updated information in public databases to ascertain relevance (oncogenicity an actionability) of detected variants. This has been clarified in the Methods, Results and Discussion sections of the manuscript.

4. CellDx alone is not sufficient to provide actionable indications for targeted therapy.

Author Response: CellDx uses updated information in public databases on actionability of variants. CellDx covers all known molecular targets in genes that are indicated for selection of FDA approved targeted therapy and immunotherapy agents in labelled setting. In addition, CellDx also provides information on variants that may be considered for selection of FDA approved targeted therapy and immunotherapy agents in off-label setting. This information has been appended to Supplementary Table S4.

5. In addition, whether and how the detected variants could be used as biomarkers to stratify patients also needs to be elaborated. 

Author Response: CellDx is a combined comprehensive Companion and Complementary Diagnostic test that can be used for (a) identification of patients who have eligible variants for selection of anticancer agents (b) identification of patients who have variants that indicate likelihood of response to anticancer agents. This has been explained in the Results and Discussion sections.

6. More evidence is needed to support the clinical significance of CellDx.

Author Response: Clinical feasibility of CellDx was explored in a preliminary study which demonstrated that CellDx can identify targetable indications (labelled and off-labeled indications) in patient samples. Among the 299 patients whose samples were evaluated, 73 (24.4%) patients were eligible for labeled treatment options, 29 (9.7%) patients were eligible for off-label treatment options and 19 (6.4%) patients were eligible for labelled as well as off-label treatment options.

Please see attachment to the decision letter for additional comments.

Excerpt 1. Page 2, paragraph 1: “CellDx includes Next Generation Sequencing (NGS) profiling of gene variants in a targeted 452-gene panel…”

1.Reviewer Comments: Please clarify the reason why selecting these 452 genes. What is their particularity? Are they known cancer genes?

Author Response: The rationale for selection of these 452 genes, the gene functions and the variant types is included in the Methods Section.

Excerpt 2. Page 4, paragraph 2: “Analytical validation was performed on 116 reference samples including Formalin Fixed Paraffin Embedded (FFPE) tumor tissue, tumor DNA or tumor RNA (Supplementary Tables S1) which were obtained from various sources such as ...”

2. Reviewer Comments: Have the genetic alterations in the 116 reference samples been orthogonally validated? If the mutation set in these samples serve as ground truth and thus are used for evaluating the sensitivity, specificity and accuracy of CellDx, are they identified by orthogonal assays? How do you get the benchmark mutation set? Please clarify.

Author Response: Analysis for Reference Samples or Sample Information Sheet for Proficiency Testing Samples. For all other types of samples, appropriate orthogonal testing was performed to ascertain the variant(s). This information has been appended to Supplementary Table S1.

3. Reviewer Comments Minor: The name of column 1 and 2 in supplementary table 1 are the same.

We regret the inadvertent error which has been rectified

Excerpt 4. Page 7, paragraph 3: “Tumor mutational burden (TMB) was determined in 133 FFPE/fresh tissue samples by NGS using ion Ampliseq 452 gene panel.”

Reviewer Comments: There are more than 133 samples involved in this study. When studying tumor mutation burden, why do you only focus on these 133 samples? Please clarify.

Author Response: NGS was performed on 133 samples, all of which were used for TMB determination. The list of sample-wise investigations is provided in Supplementary Table 2.

Excerpt 5. Page 7, paragraph 3: “TMB was defined as the total number of somatic mutations and was calculated by counting non-synonymous, coding somatic variants as well as pathogenic germline variants in tumor genome across ~2.02 MB in 452 genes.”

Reviewer Comments: Tumor mutational burden (TMB) is defined as the total number of somatic mutations and pathogenic germline variants in 452 genes. 

What are defined as pathogenic germline variants and where do you get the benchmark dataset? Is their pathogenicity specific for cancer? In some widely used datasets such as ClinVar, the mutations are responsible for different diseases. Please clarify.

The frequency of germline and somatic variants are not the same, and the two class of variants may also have distinct functional impacts in cancer. It is not fair to directly add the mutation counts.

The background mutation rates of the 452 genes could be diverse. The signal may be diluted if the mutation burden is unusually high in one or a small fraction of genes but remains the same in the other genes. Also, why is the overall mutation burden in these 452 genes informative? What is the particularity of this gene panel? Please clarify.

Author Response: We apologize for the inadvertent error in the definition of TMB which has been corrected. TMB calculation was based on somatic variants only and did not include germline variants. This addresses all other questions in the sub-section. The multi-gene NGS panel was designed to include genes with significant function and prognostic and therapeutic relevance as well as genes which were most frequently found to harbor variants in solid tumors. Thus it was sufficiently representative in terms of overall mutation coverage for determination of TMB.

Excerpt 6. Page 8, paragraph 1: “TMB levels was divided into three groups Low (1 - 5 mut/MB), Intermediate (6 - 10 mut/MB) and High (>10 mut/MB)”

Reviewer Comments: What is the basis of the arbitrary cutoff? Do you take into consideration the background mutation rate when setting the cutoff? What is the expected mutation burden? Please clarify.

Author Response: Several clinical studies have demonstrated that stratification of TMB into High, Intermediate and Low based on mut/MB was indicative for selection of, and correlated with favorable response to, immune checkpoint inhibitor therapy. These thresholds were comparable for most solid tumors with a few exceptions. Based on updated information available in literature, TMB was stratified into Low, Intermediate and High per cancer type. This has been explained in the Methods section and appropriate references have been provided. 

Excerpt 7. Page 9, paragraph 3: “A total 51 samples were used to calculate the sensitivity of CellDx test…”

Page 10, paragraph 2: “The analytical specificity of the CellDx assay was assessed for SNA, CNAs, large indels, small indels, gene fusions, MSI, PD-l and MMR, on 59 reference samples…”

Reviewer Comments: The two sample sets for assessing sensitivity and specificity are mutual exclusive. Why? Although there are some problems with the calculation of specificity (Please see Comment 10), the samples in both sets could be used to assess sensitivity. Why not merge the two sets and assess the overall sensitivity so that the result would be more comprehensive?

Author Response: As per the reviewer’s suggestion, the separate datasets for Sensitivity and Specificity have been merged and the combined dataset has been used for calculation of Sensitivity, Specificity and Accuracy.

Excerpt 8. Page 10, paragraph 2: “Additionally, 20 known negative samples for PDLI (10 for 28-8, 10 for 22C3), 8 known negatives for MMR and 2 known negative for MSI samples were included for specificity… The analytical specificity was obtained as the fraction of true negatives (TN) among the sum total of TN and false positive (FP), [TN / (TN + FP)].”

Reviewer Comments:As indicated in the manuscript, the specificity is calculated as TN / (TN + FP). How do you get the value of TN? Is it defined as the number of known negative samples where no genetic alteration is detected? If so, the unit of TN and FP will be different. The unit of FP is the number of mutations, while the unit of TN is the number of samples. Therefore, this may not be an appropriate way to assess specificity.

Author Response: Annotation of TP, TN, FP and FN are based on the presence or absence (and detectability or undetectability) of variants in the 452 genes. Thus, the definitions were:

True Positive (TP) = expected variant detected. 

False Negative (FN) = expected variant not detected. 

True Negatives (TN) = expected wild type detected (i.e., no unexpected variants detected). 

False Positive (FP) = expected wild type not detected (i.e., unexpected variant detected).

Supplementary Table S7 provides data on number of TP, FN, TN, FP which are used for calculation of Sensitivity, Specificity and Accuracy. 

Excerpt 9. Page 10, paragraph 3: “Accuracy was defined as the proportion (%) of TP and TN among the sum total of TP, TN, FP and FN.”

Reviewer Comments: The same problem as described in Comment 10. The unit of TN is different from that of TP, FP and FN. In this case it may not be an appropriate way to assess the accuracy by calculating (TP + TN) / (TP + TN + FP + FN).

Author Response: As described above.

Excerpt 10. Page 11, paragraph 1: “Reproducibility was assessed for Ampliseq 452 gene panel and PD-L1 status, whereas Repeatability was assessed for Ampliseq 452 gene panel and MSI status.”

Reviewer Comments: Why not assess reproducibility and repeatability for 452 gene panel, MSI, PDB-L1 and MMR respectively? Please clarify the reason why assessing reproducibility for gene panel and PD-L1 while assessing repeatability for gene panel and MSI.

Author Response: Repeatability as well as Reproducibility has been evaluated for all assays. The Methods and Results sections have been updated.

Excerpt 11. Page 11, paragraph 2: “The real-world performance of the CellDx assay estimated by analysing clinical samples for detection of actionable features in DNA/RNA, as well as in determining status of PD-L1, MMR, MSI and TMB.”

Reviewer Comments: The statement here is confusing. How do you estimate the clinical performance? Do you mean analyzing clinical samples using CellDx and looking into the genetic alterations detected, and see how many of them are biologically actionable? Please provide more details.

Author Response: A preliminary feasibility study was conducted with clinical samples to determine the ability of CellDx to identify variants indicative for selection of targeted or immune therapy agents in labelled and off-label setting. The statement has been restructured for clarity. 

Excerpt 12. Page 12, paragraph 2: “Among the 27 samples which were analyzed by NGS, 176 variants were detected in 65 genes including 148 SNA, 17 indels, 1 CNA and 10 gene fusions. There were 2 false negatives (FN) for which BAM files of the samples were reviewed using integrated genome viewer: one FN was due to 2% variant allele frequency (VAF) of EGFR.pL858R and one FN was due to 2.8 copies of MET. The overall analytical sensitivity for SNA, small indels, large indels, CNA and gene fusion were 99.99%...The overall sensitivity of the CellDx assay was 99.99%.”

Reviewer Comments: As indicated here, for NGS, TP = 176 and FN = 2. The analytical sensitivity should be calculated as TP / (TP + FN) = 98.9%. How was 99.99% calculated? In Supplementary Table S5, I found no false negative listed there, which is in conflict with the statement of 2 FN. Furthermore, in the “Sensitivity” section of Table S5, the first entry is “NGS: NGS: CNA” and the fifth entry is “NGS: CNA”, which seems to be duplicate. The TP value in the “Overall” entry is 51, how was 51 calculated? Why is the overall sensitivity of CellDx 99.99%? Please clarify.

Author Response: The numbers cited in the section have been reviewed and confirmed against the numbers in Supplementary Table 7 (which was previously Supplementary Table 5). Updated Supplementary Table S7 indicates the test-wise and overall number of TP, FN, TN and FP, as well as test-wise and overall Sensitivity, Specificity and Accuracy. Please refer to the FN column which shows where the FN were observed. Overall TP for NGS is the sum of TP for SNV, CNA, Indels and Fusions. Test-wise and overall Analytical Sensitivities have been recalculated. The duplicate entries have been resolved. The data tables have been combined for better clarity. 

Excerpt 13. Page 13, paragraph 1: “…1078 true negative (TN) variants were correctly detected.”

Reviewer Comments: How are the TN variants defined? If they are not known variants and are not detected by NGS, how do you get the variant list? Also, I couldn’t get this number “1078” by summing the NGS entries in the “Specificity” section in Supplementary Table S5. In this table section, again, what do “NGS: NGS: CNA” and “Overall” mean? Please clarify.

Author Response: As described in a prior response, True Negatives (TN) = expected wild type detected (no unexpected variants detected). Since the Sensitivity and Specificity numbers have been merged, please refer to updated Supplementary Table S7 for numbers of TP, FN, TN, FP for calculation of Sensitivity, Specificity and Accuracy.

Supplementary Table S7 also includes the definitions of all acronyms.

Excerpt 14. Page 13, paragraph 2: “Accuracy of NGS based on detection of all TP variants and 2 FP among all TN variants was 99.9%. Accuracies for PD-L1, MMR and MSI were also be 99.99%. The overall accuracy of the CellDx assay was 99.32% (95% CI: 98.71% - 99.69%).”

Reviewer Comments: As the questions remain to be answered on the calculation of sensitivity and specificity, the validity of this accuracy is unclear.

Author Response: As suggested by the reviewer, Sensitivity and Specificity datasets have been merged and these parameters as well as Accuracy have been recalculated. Please refer to updated Supplementary Table S7 for numbers of TP, FN, TN, FP as well as the calculated Sensitivity, Specificity and Accuracy.

Excerpt 15. Page 13, paragraph 3: “Repeatability was determined for NGS and MSI…Reproducibility was determined for the NGS and PD-L1.”

Reviewer Comments: Why do you only assess the repeatability of NGS and MSI? Why not assess the repeatability of MMR and PD-L1? 

Reviewer Comments: Why only assess the reproducibility of NGS and PD-L1? Why not assess the reproducibility of MSI and MMR? 

Reviewer Comments: What is the definition of precision? How do you integrate repeatability and reproducibility to get the precision value? Please clarify.

Author Response: 15, 16. We have incorporated data on evaluation of Repeatability as well as Reproducibility for all assays. The relevant sections in Methods and Results have been updated. 

Author Response: 17. Precision was defined as the concordance between replicate assays with the same sample. Replicate assays were performed by the same analyst (Intra-Operator) to determine Repeatability, and by different analysts (Inter-Operator) to determine Reproducibility. Replicate datasets used for assessing Repeatability and Reproducibility were merged and evaluated for overall concordance, i.e., Precision. 

Excerpt 16. Page 14, paragraph 2: “To evaluate the LOD of variant calling, 2 FFPE tissue samples were analysed;”

Reviewer Comments: It seems that the LOD assessed from 2 samples may not be solid enough.

Author Response: LOD was assessed over a range of dilutions for each of 4 well characterized variants, including one CNA, one SNV, one InDel and one Fusion. One reference sample was used for evaluating 3 variants and another reference sample was used for evaluating the 4th variant type. Updated Supplementary Table S5 represents the findings with more clarity.

Excerpt 17. LOD was assessed over a range of dilutions for each of 4 well characterized variants, including one CNA, one SNV, one InDel and one Fusion. One reference sample was used for evaluating 3 variants and another reference sample was used for evaluating the 4th variant type. Updated Supplementary Table S5 represents the findings with more clarity.

Reviewer Comments: What is the source of pathogenic and likely pathogenic somatic variants? How did you get the benchmark dataset? Are these variants causal in cancer? 

Reviewer Comments: Where do you get the VUS list? The function of VUS is unknown. By stating that these VUS together with pathogenic variants were detected in 96.99% samples, what conclusion do you want to make? 

Author Response: 19. The variant calling file was queried using Ingenuity software version 5.6 (Qiagen, Germany) and PredictSNP2 to annotate the variants (Pathogenic / Likely-Pathogenic / Driver / Passenger / VUS) based on updated information in public databases – which has been explained in the Methods section. 

Author Response: 20. Information on variants annotated as VUS is obtained from databases indicated in the Methods section. Pre-emptive reporting of VUS may guide future therapeutic decisions as and when prognostic or therapeutic relevance of the VUS is established.

Excerpt 18. Page 15, paragraph 2: “We analyzed the clinical associations of all detected variants in all patients by mapping driver mutations to clinical annotation databases (CIViC and OncoKB). Among the 133 patients’ samples, 852 unique mutations were detected, of which 784 were oncogenic and 47 were actionable. ”

Reviewer Comments: All detected variants are called “driver mutations” here, which is not rigorous. How do you know they are all oncogenic? 

Reviewer Comments: The oncogenicity of the detected variants is annotated by external databases. CellDx itself is not able to identify either oncogenic or actionable genetic alterations.

Author Response: We regret the inadvertent error in framing the statement. The statement has been reworded for better clarity. CellDx is not intended for functional annotation of detected variants. CellDx retrieves updated information in public databases to report oncogenicity / actionability of detected variants.

Excerpt 19. Page 15, paragraph 2: “Among the 128 patients, 31 patients (27%) had low TMB, 50 patients (43%) had an intermediate TMB and 34 patients (30%) had high TMB…Among the 112 samples evaluated by IHC for PD-L1 expression, 22 samples (19.64%) were positive (TPS score >1%) for PD-359 L1. Among the 105 samples evaluated for MMR status, LNE: MSH2 + MSH6 was observed in 1 sample. Among the 138 samples evaluated for MSI status, 6 samples (4%) returned positive findings with BAT-26 (5%), NR-21 (2%), NR-24(2%) and MONO-27 (1%).”

Reviewer Comments: How do TMB, PD-L1 expression, MMR status, and MSI status stratify patients? What does it mean if a patient has low, intermediate or high TMB? What does it mean if a sample is positive for PD-L1 or MSI? Does that mean worse clinical outcome? What is the clinical significance of studying TMB, PD-L1 expression, MMR status, and MSI status? Please clarify.

Author Response: Status of TMB, PD-L1, MMR and MSI are relevant for selection of immune checkpoint inhibitor therapy agents and identify patients who are likely to respond to these agents. Low, Intermediate and High TMB, are associated with increasing probability of response to Immunotherapy agents. Similarly, MSI-high and PD-L1 positivity are associated with response to Immunotherapy agents. These points have been explained in the results and discussion section.

---

## [Decision Letter · Decision Letter 1]

4 Jan 2021

PONE-D-20-23324R1

Development and Validation of a Multigene Variant Profiling Assay to Guide Targeted and Immuno Therapy Selection in Solid tumors

PLOS ONE

Dear Dr. Akolkar,

Thank you for submitting your manuscript to PLOS ONE. After careful consideration, we feel that it has merit but does not fully meet PLOS ONE’s publication criteria as it currently stands. Therefore, we invite you to submit a revised version of the manuscript that addresses the points raised during the review process.

Reviewer 1 has a highlighted a few remaining minor issues that require further clarification and/or minor edits to the manuscript. 

We look forward to receiving your revised manuscript.

Kind regards,

Robert S. Weiss

Academic Editor

PLOS ONE

Reviewers' comments:

Reviewer's Responses to Questions

**Comments to the Author**

1. If the authors have adequately addressed your comments raised in a previous round of review and you feel that this manuscript is now acceptable for publication, you may indicate that here to bypass the “Comments to the Author” section, enter your conflict of interest statement in the “Confidential to Editor” section, and submit your "Accept" recommendation.

Reviewer #1: (No Response)

Reviewer #2: All comments have been addressed

2. Is the manuscript technically sound, and do the data support the conclusions?

Reviewer #1: Yes

Reviewer #2: Yes

3. Has the statistical analysis been performed appropriately and rigorously? 

Reviewer #1: No

Reviewer #2: Yes

4. Have the authors made all data underlying the findings in their manuscript fully available?

Reviewer #1: Yes

Reviewer #2: Yes

5. Is the manuscript presented in an intelligible fashion and written in standard English?

Reviewer #1: Yes

Reviewer #2: Yes

6. Review Comments to the Author

Reviewer #1: In Supplementary Table S7 of the revision, for NGS:SNA, the numbers for TP, FP, TN and FN are 148, 9, 1053 and 1, respectively. However, in the Supplementary Table S5 of the previous version, the number for FN was 0. Please clarify what re-analyses have led to this big change in FN, although the absolute number being only 1.

I suggest to move the four numbers used for calculating each statistic in Table 1 from the supplementary table to Table 1. This will help readers understand the underlying sample sizes, although the 95% CIs could hint on how big the underlying numbers are.

In Supplementary Table S5, it is titled "S7". It is unclear what "(Mean depth 500x, >99.1% coverage >100X)" in the title means. After reading the manuscript multiple times, I see the authors meant they aimed at a mean depth of 500X, which was not a result data of the study. Please consider to remove from Table S5.

Reviewer #2: The authors have addressed all my comments in the revised manuscript. I do not have any more comments.

7. PLOS authors have the option to publish the peer review history of their article (what does this mean?). If published, this will include your full peer review and any attached files.

Reviewer #1: No

Reviewer #2: No

---

## [Author Response · Author response to Decision Letter 1]

7 Jan 2021

QUERY 1: In Supplementary Table S7 of the revision, for NGS:SNA, the numbers for TP, FP, TN and FN are 148, 9, 1053 and 1, respectively. However, in the Supplementary Table S5 of the previous version, the number for FN was 0. Please clarify what re-analyses have led to this big change in FN, although the absolute number being only 1.

RESPONSE: In the version submitted initially, Sensitivity and Specificity were evaluated in separate sample sets. The Sensitivity sample set (n = 51) was evaluated only for TP and FN only (TN and FP not required for Sensitivity). The Specificity sample set (n = 59) was evaluated for TN and FP (TP and FN not required for Specificity). Both these sample sets were merged as suggested by the Reviewer, and all samples in the combined sample set (n = 110) were evaluated for TP, TN, FP and FN to derive Sensitivity and Specificity. The FN in question was reported in a sample which was previously evaluated for TN and FP only (erstwhile Specificity set). Thus, this FN value was not reported in the prior Supplementary Table S5, but is now reported in Supplementary Table S7.

QUERY 2: I suggest to move the four numbers used for calculating each statistic in Table 1 from the supplementary table to Table 1. This will help readers understand the underlying sample sizes, although the 95% CIs could hint on how big the underlying numbers are.

RESPONSE: As the Reviewer rightfully suggests, the 95% CI values serve to convey the significance of the underlying sample (+variant) numbers. Since Table 1 is intended to provide a snapshot of the CellDx performance characteristics, incorporating numbers for FP, FN, TP and TN might render the Table complex. However, in appreciation of the point raised by the Reviewer, Table 1 Legend has been updated to explicitly mention that FP, FN, TP, TN numbers are provided in Supplementary Table S7 for benefit of interested readers. 

QUERY 3: In Supplementary Table S5, it is titled "S7". It is unclear what "(Mean depth 500x, >99.1% coverage >100X)" in the title means. After reading the manuscript multiple times, I see the authors meant they aimed at a mean depth of 500X, which was not a result data of the study. Please consider to remove from Table S5.

RESPONSE: The Legend for Supplementary Table S5 has been revised to exclude the indicated text; the parameters indicated in this sentence are described in the Methods (Sequencing Data Analysis) and Results (NGS Data Analysis) Sections.

---

## [Editor Report · Decision Letter 2]

13 Jan 2021

Development and Validation of a Multigene Variant Profiling Assay to Guide Targeted and Immuno Therapy Selection in Solid tumors

PONE-D-20-23324R2

Dear Dr. Akolkar,

We’re pleased to inform you that your manuscript has been judged scientifically suitable for publication and will be formally accepted for publication once it meets all outstanding technical requirements.

Kind regards,

Robert S. Weiss

Academic Editor

PLOS ONE

Additional Editor Comments (optional):

The additional comments from Reviewer 1 regarding the revised manuscript have now been fully addressed.

---

## [Editor Report · Acceptance letter]

28 Jan 2021

PONE-D-20-23324R2 

Development and Validation of a Multigene Variant Profiling Assay to Guide Targeted and Immuno Therapy Selection in Solid tumors 

Dear Dr. Akolkar:

I'm pleased to inform you that your manuscript has been deemed suitable for publication in PLOS ONE. Congratulations! Your manuscript is now with our production department. 

Kind regards, 

on behalf of

Dr. Robert S. Weiss 

Academic Editor

PLOS ONE